# Characterization of tumor evolution by functional clonality and phylogenetics in hepatocellular carcinoma
Zeynep Kacar [1,2], Eric Slud [2], Doron Levy[2], Julián Candia [3], Anuradha Budhu[1,4], Marshonna Forgues[1], Xiaolin Wu [5], Arati Raziuddin [5], Bao Tran[5], Jyoti Shetty[5], Yotsawat Pomyen[6], Jittiporn Chaisaingmongkol [6], Siritida Rabibhadana[6], Benjarath Pupacdi[6], Vajarabhongsa Bhudhisawasdi[7], Nirush Lertprasertsuke[8], Chirayu Auewarakul[9], Suleeporn Sangrajrang[10], Chulabhorn Mahidol[6], Mathuros Ruchirawat [6,11] & Xin Wei Wang [1,4] ✉

Hepatocellular carcinoma (HCC) is a molecularly heterogeneous solid malignancy, and its fitness may be shaped by how its tumor cells evolve. However, ability to monitor tumor cell evolution is hampered by the presence of numerous passenger mutations that do not provide any biological consequences. Here we develop a strategy to determine the tumor clonality of three independent HCC cohorts of 524 patients with diverse etiologies and race/ethnicity by utilizing somatic mutations in cancer driver genes. We identify two main types of tumor evolution, i.e., linear, and non-linear models where non-linear type could be further divided into classes, which we call shallow branching and deep branching. We find that linear evolving HCC is less aggressive than other types. *GTF2IRD2B* mutations are enriched in HCC with linear evolution, while *TP53* mutations are the most frequent genetic alterations in HCC with non-linear models. Furthermore, we observe significant B cell enrichment in linear trees compared to non-linear trees suggesting the need for further research to uncover potential variations in immune cell types within genomically determined phylogeny types. These results hint at the possibility that tumor cells and their microenvironment may collectively influence the tumor evolution process.

Tumor cell evolution plays a key role responsible for tumor initiation and progression linked to treatment response. Cancer is a result of irreversible changes to critical genes responsible for hallmarks of cancer[1]. Somatic mutations, which are genetic changes acquired during an individual's lifetime, are frequently detected in tumor cells, and can play a driving role in the initiation and progression of tumorigenesis. These somatic mutations can be valuable in delineating the initiation and progression of tumor cells, as well as in characterizing tumor cell clonality. However, one key step in accurately defining tumor cell clonality and its evolution has been complicated by the presence of a limited number of functional mutations from a large number

of passenger mutations that make little or no contribution to tumor progression[2]. Another key consideration is how tumor evolution is shaped by the interactions between tumor cells and their microenvironment. Each tumor may contain multiple subclones with distinct phylogenic relationship and unique tumor microenvironment that provides survival benefit. Therefore, finding distinct phylogenetic features and underlying molecular influences that shape tumor clonal phylogenies and consequently drive tumor evolution would improve our understanding of tumor clonality and shed light on the possibility of personalized cancer medicine for solid tumors. One strategy to overcome this challenge is to use somatic mutations

[1]Laboratory of Human Carcinogenesis, Center for Cancer Research, National Cancer Institute, Bethesda, MD 20892, USA. [2]Department of Mathematics, University of Maryland, College Park, MD 20742, USA. [3]Longitudinal Studies Section, Translational Gerontology Branch, National Institute on Aging, Baltimore, MD 21224, USA. [4]Liver Cancer Program, Center for Cancer Research, National Cancer Institute, Bethesda, MD 20892, USA. [5]Cancer Research Technology Program, Frederick, MD 21702, USA. [6]Laboratory of Chemical Carcinogenesis, Chulabhorn Research Institute, Bangkok 10210, Thailand. [7]Faculty of Medicine, Khon Kaen University, Khon Kaen 40002, Thailand. [8]Faculty of Medicine, Chiang Mai University, Chiang Mai 50200, Thailand. [9]Princess Srisavangavadhana College of Medicine, Chulabhorn Royal Academy, Bangkok 10210, Thailand. [10]National Cancer Institute, Bangkok 10400, Thailand. [11]Center of Excellence on Environmental Health and Toxicology (EHT), OPS, MHESI, Bangkok, Thailand. ✉e-mail: xw3u@nih.gov

found in known cancer driver genes that may help in improving the accuracy of tumor clonality analysis and in studying the biological nature of clonal evolution.

Worldwide, more than one million people are diagnosed with liver cancer each year. Hepatocellular carcinoma (HCC) is the most common type of liver cancer, with poor prognosis even at an early stage[3]. HCC exhibits numerous distinct biological and molecular features within the same tumor known as intratumor heterogeneity (ITH)[4]. There are several potential underlying mechanisms driving ITH in HCC tumors[5], including driver mutations, adaptive tumor microenvironment, and various mutational signature profiles reflected from complex etiological exposures. In this work, we aim to reconstruct tumor clonal trees of HCC tumors using whole exome sequencing data derived from three independent cohorts of 524 patients with diverse race/ethnicity and etiologies, i.e., the TIGER-LC cohort, NCI-MONGOLIA cohort, and the TCGA-LIHC cohort[6–8]. Our data suggest that there are three distinct groups of clonal trees within HCC, which we call linear, shallow branching, and deep branching based on the properties of these distinct clusters of trees. We identify key potential drivers for each phylogeny type by profiling the potential clonal mutations of phylogenetic tumor trees. Additionally, our findings suggest that integrating clonality analysis with immune cell decomposition data (from RNA-sequencing data) could provide key insights into the distinct interactions of these phylogenetic tree classes with the tumor microenvironment.

## Results

### Reconstruction of tumor clonal trees

Each patient's tumor consists of varying proportions of subclones due to tumor cell evolution. To uncover these clones' ancestral relationship in HCC cohorts, we reconstructed phylogenetic trees of tumor samples using mutation read-count data from Whole Exome DNA sequencing while focusing on the functional mutations occurring on liver or liver cancer related genes. Using a frequentist approach, SMASH (Subclone Multiplicity Allocation and Somatic Heterogeneity)[9], we clustered somatic mutations accounting for their copy number alteration estimates to identify sub-clones in tumor samples. For all tree topology configurations consisting of 1–5 subclones, maximum likelihood for the SMASH model parameters was calculated, leading to inferences of the most likely tree configuration. In this study, clonality and clustering analyses (Fig. 1) were implemented for three independent HCC cohorts, namely, TIGER-LC[6], NCI-MONGOLIA[7], and TCGA-LIHC[8] whose characteristics are given in Supplementary Table 1.

### Functional clonality reveals the differences between linear and nonlinear tumor evolution

Since not all gene mutations play a functional role in tumor evolution[10], we targeted a subset of gene mutations that may be likely players in liver tumor progression. Specifically, we selected 1006 functional genes, 386 of which are over-expressed in normal liver[11,12] and the remaining ones are candidate drivers of liver cancer that were studied previously (Supplementary Data 1). Our study gave particular attention to the TIGER-LC cohort due to its unique status as an unpublished dataset at the time of our analysis. As such, we initiated the "all mutations versus functional mutations" analysis within this cohort. However, recognizing the advantages of a larger sample size, we also integrated the TCGA-LIHC cohort (with $n = 375$) into our analysis. It is worth noting that the NCI-MONGOLIA cohort exhibited distinct demographic characteristics compared to the other two cohorts (Supplementary Table 2), and its patient count was significantly lower at 71 in comparison to the TCGA-LIHC cohort. Consequently, our comparative analysis was mainly centered on the TCGA-LIHC and TIGER-LC cohorts. This approach allowed us to perform a comprehensive clonality analysis that encompassed both functional gene mutations and all identified mutations. As shown in Fig. 1b, we compared the survival curves between two types of clonal models, i.e., linear and nonlinear models, resulting from the two analyses for the TCGA-LIHC cohort and found that the estimated median survivals were significantly different for the functional-mutation clonality analysis but not for the analysis of all mutations. Our main hypothesis to test in this study is of no survival difference between linear versus nonlinear tree types in all cohorts versus alternatives where all cohorts show linear versus nonlinear effects in the same direction. To test this, we specifically calculated a cohort stratified log-rank test. Strikingly, this test unveiled a highly significant $p$-value of 0.00001, indicating a strong association between the tree-type variable (linear or nonlinear) and survival (Supplementary Table 3). Furthermore, we provided TIGER-LC cohort specific log-rank test $p$-values

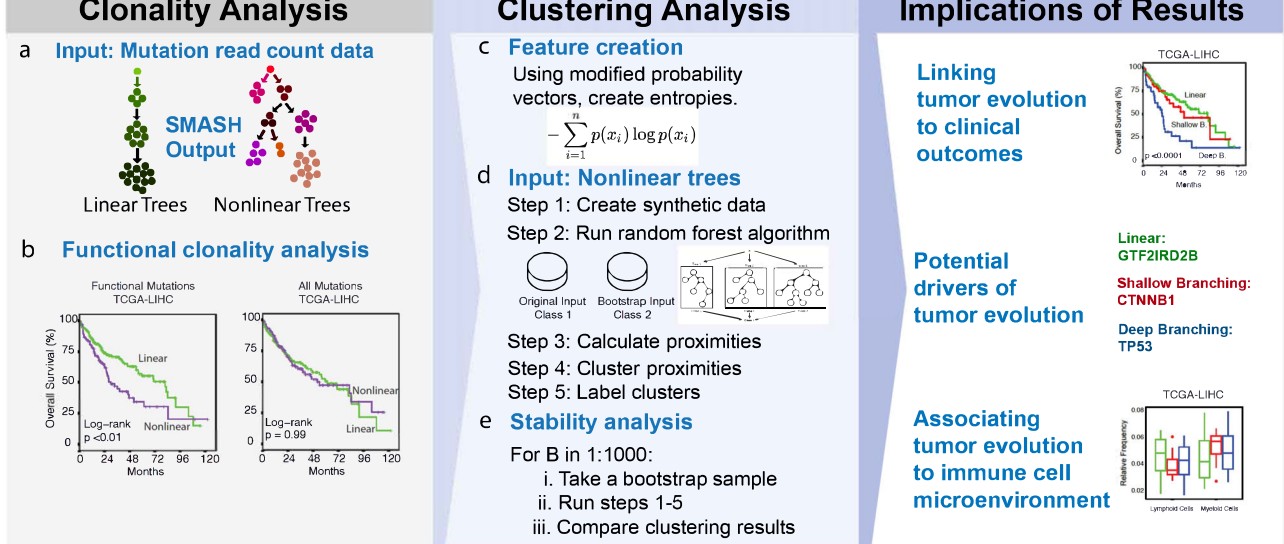

**Fig. 1 | Clonality analysis of hepatocellular carcinoma.** First panel summarizes the clonality analysis: **a** The SMASH takes mutation read count data as an input and outputs linear and non-linear trees. We further selected functional gene mutations to do the clonality analysis. **b** Overall survival of patients from linear and nonlinear phylogenies for TCGA-LIHC cohort for functional mutations (on the left) and for all mutations (on the right). Linear and nonlinear phylogenies were determined based on the SMASH results. Second panel summarizes the clustering analysis: **c** Features were created by using modified probability vectors from SMASH outputs. **d** Using nonlinear trees with created features, clustering algorithm was implemented to find shallow branching and deep branching tree groups. **e** Stability analysis was conducted to make sure well-separated and stable clusters obtained. Final panel summarizes the implications of the results on clinical outcomes.

in Supplementary Fig. 1. The forest plots in Supplementary Fig. 2 further illustrate this finding, as significant hazard ratios with 95% confidence intervals were observed in the models with functional mutations but not in those with all mutations, for both cohorts. These results suggest that targeting functional mutations may be a more effective strategy that targeting all mutations in identifying patient subgroups with different survival.

## Classification of nonlinear tumor trees into shallow and deep branching phylogenies

In the literature, two common types of nonlinear evolution models have been proposed based on the presence or absence of selection advantages among subclones[13]. To distinguish distinct tree clusters within the nonlinear class, we developed a clustering algorithm that initially created several diversity variables (entropies) for each tree using subclone proportions and proportions of mutations by subclone and then used these created variables to cluster and label trees based on how balanced they were, i.e., based on how similar the sizes and numbers of mutations in the subclones were. We classified nonlinear trees as shallow branching or deep branching, depending on their structure. Branching trees with similarly sized branches and subclones were classified as deep branching, and those without this feature were designated shallow branching. The details of the clustering algorithm are outlined in the Methods section.

Supplementary Figs. 3–5 depict scatter plots of the generated diversity variables with the colors representing clustering assignments (red: shallow branching, black: deep branching) for nonlinear trees coming from the SMASH algorithm for all three cohorts. These plots confirm the separation achieved by clustering and the importance of the generated variables in distinguishing shallow branching and deep branching tumor trees. Furthermore, Supplementary Fig. 6 demonstrates the clear separation of these two tree clusters by displaying PC2 versus PC1 plots for the TCGA-LIHC and TIGER-LC cohorts and the PC3 versus PC1 plot for the NCI-MONGOLIA cohort. All three plots are color-coded according to the shallow and deep branching clusters. The same features used in Supplementary Figs. 3–5 were utilized for PC displays. Supplementary Table 4 provides log-rank test p-values, showcasing the success of our clustering algorithm in robustly distinguishing between shallow and deep branching trees. While the contrasts for the single NCI-MONGOLIA cohort (as indicated in Supplementary Table 4) are not statistically significant, Supplementary Table 3 highlights a significant difference in survival outcomes when comparing shallow branching and deep branching trees stratified by cohort. Furthermore, when examining the linear versus shallow branching tree types, we did not observe a statistically significant disparity in survival outcomes. However, linear trees displayed a notably lower total mutation count compared to their shallow branching counterparts, as is evident in Supplementary Table 5. These findings collectively support the distinctions between various phylogeny types and their potential implications for survival and genetic characteristics in our study.

## Phylogenetic trees of HCC cohorts

The clonality and clustering analysis of this study revealed the existence of three distinct phylogenetic tree groups within HCC cohorts: linear, shallow branching, and deep branching. Figure 2a depicts a linear tree from the TCGA-LIHC cohort (analysis with functional mutations) where the founding clone accounts for 61% of tumor cells and harbors two mutations. Subsequently, mutations accumulate sequentially, resulting in the formation of subclones 1, 2, and 3. The numbers adjacent to the branches indicate the number of unique mutations in the corresponding subclone, and the branch lengths are proportional to this number.

Figure 2b, c displays two types of branching trees observed in the TCGA-LIHC cohort (analysis with functional mutations). Shallow branching and deep branching trees differ in that deep branching trees show a relatively high degree of balance with respect to edge-lengths (numbers of new mutations) and prevalence (read-counts). In Fig. 2b, c, the proportion entropy (PE) and the mutation entropy (ME) were both 1.14 for shallow branching tree, whereas for the deep branching tree PE was 1.45 and ME was 1.57.

Our clonality analysis results indicate that TCGA-LIHC cohort consisted mostly of linear trees (270 linear, 56 shallow branching, 49 deep branching), while the TIGER-LC (22 linear, 32 shallow branching, 24 deep branching) and NCI-MONGOLIA (18 linear, 18 shallow branching, 35 deep branching) cohorts had a higher proportion of shallow and deep branching trees.

## Stability analysis of the algorithm identifying shallow and deep branching trees

In this work, we devised a clustering and labeling algorithm to assign a shallow branching or deep branching label for subjects with nonlinear clonal trees as determined by the SMASH algorithm. Figure 1c, d summarizes the steps we followed for the purpose of clustering. For any clustering analysis, it is vital that the clusters were constructed in a stable manner because unstable clustering results might invalidate our biological interpretation[14]. To ensure the robustness of the clustering, we performed a stability analysis (Fig. 1e) to demonstrate that our algorithm produces stable clustering results under bootstrap resampling. Following clustering, we applied a labeling algorithm to assign a shallow branching or deep branching label to each tree based on its balance. Further details on this process are provided in the Methods section. To assess the stability of the clustering results, we aligned the original and bootstrap clustering outcomes, which had already been labeled as shallow branching or deep branching, to construct a 2 × 2 frequency table. Using this table, we computed a similarity score[15] for each bootstrap result. The estimated average log odds-ratios were found to be 5.45 for TCGA-LHC, 6.02 for TIGER-LC, and 6.64 for NCI-MONGOLIA, indicating high agreement between cluster results.

## Linking tumor evolution to clinical outcomes

To investigate the impact of tumor evolution on survival, we examined the Kaplan Meier survival plots for groups of subjects defined by phylogenetic clusters. We found that overall survival for trees that are linear have statistically better prognosis compared to the non-linear ones (as shown in Fig. 2d–f). In addition, as expected, we observed that non-linear trees had significantly higher ITH compared to linear trees (Fig. 2g–i). Since we labeled the deep branching trees as the ones that have higher diversity among non-linear ones, this was already true for deep branching being the most diverse (high ITH). We also saw that linear trees had the lowest ITH measured by the created entropy variables.

We explored a potential correlation between tumor phylogeny type and stage, despite encountering challenges related to the substantial absence of the cancer stage variable in the NCI-MONGOLIA and TIGER-LIHC cohorts. Our findings, outlined in Supplementary Tables 6–8, are based on a test for the null hypothesis of row-column independence within each cohort. Notably, our analysis did not reveal a distinct association between phylogeny type and stage.

To explore the driving factors behind the observed association suggesting heightened aggressiveness of deep branching tumors, we also aimed to identify suggestive evidence of variables significantly impacting survival, with a primary focus on the Tree-type variable significance. For this purpose, we conducted multivariate survival analysis employing the Cox Proportional-Hazards Regression framework. However, it's crucial to acknowledge the potential limitations in drawing conclusions related to the Stage variable due to our reliance on complete-case analysis and the exceptionally high, unexplained missing rates for Stage in the TIGER-LC and NCI-MONGOLIA cohorts. The results are summarized in the Analysis of Deviance Tables (Supplementary Tables 9–12) across all cohorts, reflecting the outcomes of our multivariate Cox proportional hazards models.

Copy number alteration is one of the factors that complicate clonality analysis. Therefore, we estimated copy number alterations prior to reconstructing clonal trees to ensure accurate results. Using paired tumor-non tumor whole exome sequencing data, we estimated ploidy and allele specific copy number alterations of our samples using Sequenza[16]. We conducted a comparative analysis of ploidy estimates across various tree phylogenies, and the detailed results of this

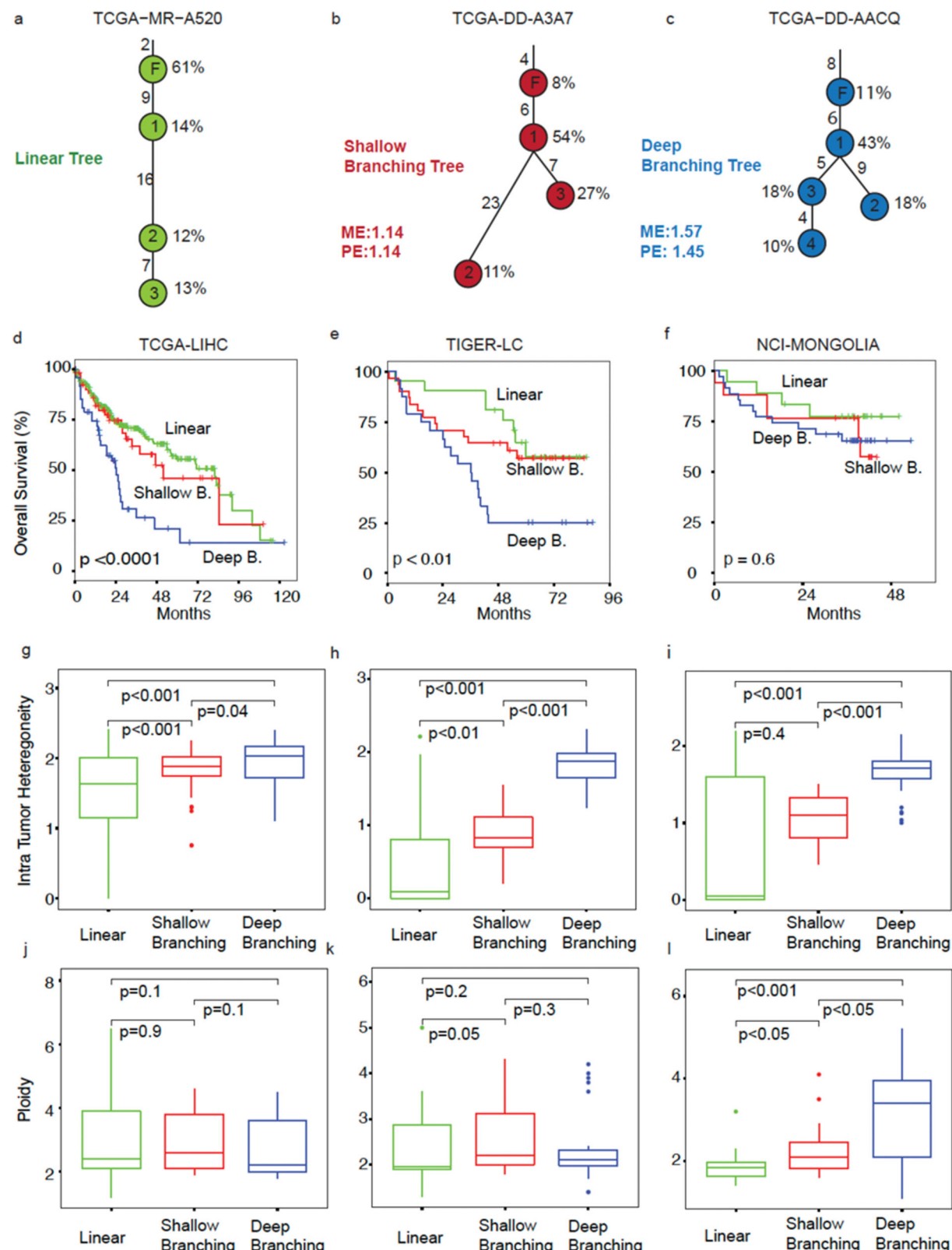

comparison are given in Fig. 2j–l. The box plots visually represent these findings, and we have included the *p*-values for 2 sample t-tests of pairwise comparisons. Notably, our analysis revealed a statistically significant difference in ploidy estimates between different tree phylogenies, but this distinction was observed exclusively within the NCI-MONGOLIA cohort.

## Potential drivers of linear, shallow, and deep branching phylogenies of HCC

One possible explanation for the diverse paths of tumor evolution observed in HCC tumors could be the presence of distinct driver mutations for each type of tumor evolution phylogeny. To explore this hypothesis, we compared the mutation profiles of linear, shallow branching, and deep

**Fig. 2 | Linking tumor evolution to clinical outcomes. a–c** Three representative samples from TCGA-LIHC cohort showing linear, shallow branching, and deep branching tree evolution models. Percentages next to the nodes (clones) show the subclone proportion of each clone. Numbers next to the branches represent the number of unique mutations for the corresponding clone. The F inside the nodes stands for the founding clone and the numbers inside the nodes stand for the subclone indices. Mutation entropy (ME) and proportion entropy (PE) (see methods) values for shallow branching and deep branching trees were also provided. **d–f** Overall survival plots of samples grouped as linear, shallow branching, and deep branching for the cohorts TCGA-LIHC, TIGER-LC and NCI-MONGOLIA respectively. P-values are calculated using log-rank test statistics for three groups. **g–i** Intra tumor heterogeneity measured by product of ME and PE versus tree evolution models for the cohorts TCGA-LIHC, TIGER-LC, and NCI-MONGOLIA. **j–l** Box plots illustrating the distribution of ploidy estimates across different phylogeny types within the cohorts. The p-values provided in the boxplots are pairwise mean comparison test p-values. Analysis of variance (ANOVA) conducted on these data revealed statistically significant differences in ploidy estimates among the various phylogeny types within the NCI-MONGOLIA cohort.

branching phylogenies. We defined a gene as a potential driver among all the functional genes we analyzed if it was mutated in the founding clone.

Figure 3 presents potential driver profiles of linear, shallow branching, and deep branching trees in TCGA-LIHC (top row), TIGER-LC (second row), and NCI-MONGOLIA (third row) cohorts. In all cohorts, deep branching trees exhibited the highest rate of driver mutations in *TP53*, whereas in linear trees, *TP53* was not the most mutated gene. Notably, for non-linear phylogenies, *TP53* and *CTNNB1* were the two most frequently mutated genes, whereas for linear phylogenies, they did not even rank within the top 5 in the TIGER-LC and NCI-MONGOLIA cohorts. In the TCGA-LIHC cohort, these gene mutations remained dominant in non-linear trees, but in linear trees, *MUC6* was the most frequently mutated gene across all patients. While this trend was observed, no significant correlation was found between *TP53* mutation and phylogenetic model evolution. Conversely, the mutation frequencies of the *GTF2IRD2B* gene showed an opposite trend, suggesting that it may be a driver of linear evolution, particularly in the NCI-MONGOLIA cohort. To validate this, we performed the Freeman-Halton extension[17] of the Fisher's exact test for $3 \times 3$ contingency tables to check the association between mutation status of *GTF2IRD2B* (no mutation, driver, not driver) and tree evolution model (linear, shallow branching, deep branching). In all cohorts, the exact test p-values (<0.01) indicated that the *GTF2IRD2B* gene status was associated with the tree evolution phylogeny, suggesting its potential role as a driver for linear phylogenies. To further investigate hotspot mutations in this gene, we examined the lollipop plot (Fig. 3j) and identified a hotspot mutation that occurred 201 times across all cohorts. Although our p-values did not provide compelling evidence supporting *TP53* as a driver of non-linear trees, we examined the mutation locations of this critical gene in HCC. We discovered that the majority of *TP53* mutations occurred in the DNA binding domain, but we also identified 5 instances in the *p53* tetramerization domain, which is known to impact *p53* transcriptional activity (Fig. 3k).

## Immune cell microenvironment of tumor phylogenies of HCC

The interplay between tumor cells and their microenvironment is critical in tumor progression and treatment response. It has been demonstrated that the clonal architecture of tumors can shape their microenvironment[18]. To investigate this further, we utilized RNA-sequencing data from the same tumor samples and performed a transcriptome analysis using CIBER-SORTx, a deconvolution tool[19].

We first looked at B cells specifically because B cells play an important role in the immune response against cancer. Our results, as depicted in Fig. 4a–c, demonstrate that the total B cell frequencies are significantly higher in linear trees compared to non-linear trees for all cohorts (Kruskal–Wallis's test p-values ≤ 0.05). Additionally, the stacked bar charts in Fig. 4d show the mean relative frequencies of immune cell types for linear, shallow branching, and deep branching trees in the NCI-MONGOLIA cohort.

Examining immune cell type frequencies for all three cohorts (Supplementary Fig. 7), we observed a slight trend suggesting myeloid and lymphoid cells may impact the evolution of clonal trees in opposite directions. Although further investigation of myeloid and lymphoid cell frequencies by tumor evolution model (Fig. 4e–g) did not reveal statistically significant results, we observed for all three cohorts' shallow branching trees had the highest average amount of myeloid cell frequencies and the lowest average amount of lymphoid cell frequencies compared to linear. These

findings suggest that tumor evolution models may impact the relative abundance and distribution of immune cell types within the tumor microenvironment.

## Mutational signatures of tumor phylogenies of HCC

Another underlying mechanism that could explain the different evolutionary models is mutational signature exposures or the imprints of somatic alterations in the genome caused by carcinogenic exposures or environmental factors. To investigate this within our HCC cohorts, we analyzed mutational signatures that consist of frequency patterns among 96 trinucleotides, formed by enumerating all single-nucleotide combinations before and after each one of 6 possible single nucleotide substitutions. After excluding the signatures that are all zero for all patients in the cohort, we ended up with 43 signatures for TCGA-LIHC, 15 signatures for TIGER-LC and 14 signatures for NCI-MONGOLIA. Multi-panel bar plots in Supplementary Fig. 8 show the relative exposures of COSMIC[20,21] reference mutational signatures for HCC cohorts. TIGER-LC and NCI-MONGOLIA cohorts showed similar mutational exposure profiles compared to TCGA-LIHC.

However, we did not observe any statistically significant differences in mutational exposure profiles between linear and non-linear trees. These findings suggest that mutational signature exposures may not be the primary driver of the observed differences in evolutionary models among HCC cohorts. Further investigation is required to identify other underlying mechanisms responsible for the observed differences in HCC evolutionary models.

## Discussion

Intra-tumor heterogeneity (ITH) is the phenomenon of clonal variability within a patient's tumor, which arises as a result of stochastic (mutation, drift) and deterministic (selection) processes in the evolution of cancer[22]. A typical tumor evolution analysis combines single nucleotide variants (SNVs) and copy number alterations (CNAs)[9,23–25], but not all SNVs are relevant to cancer progression. Moreover, many phylogenetic reconstruction algorithms are unreliable when the number of mutations is high. To deal with redundancy in mutations, we focus on liver and liver cancer specific genes, successfully showing that functional mutations improve clonality analysis.

Another challenge of clonality analysis is that there is no consensus optimal approach for ITH inference and for quantifying ITH in the literature. ITH quantifications that were made from previous studies include but are not limited to clone counts[26], mutant allele tumor heterogeneity (MATH)[27], and classic or proportion entropy[9]. In this study, we represented ITH by reconstructing the clonal trees and calculating various features using mutation entropy as well as the proportion entropy that previous studies used. By combining these representations of ITH with phylogenetic tree construction, we successfully compared the biological and clinical paths of tumor evolution.

The first and most well-known model of tumor evolution is the linear evolution model based on the ideas of Nowell[28], where tumors accumulate clonal mutations with highly dominant selective properties, outcompeting all previous clones. For a long time, tumor evolution was only believed to be a linear accumulation of clonal mutations. However, observations in several studies[29,30] showed the possibility of nonlinear growth for tumors with several molecularly distinct subclones. The SMASH algorithm[9] identifies linear or nonlinear trees, where linear trees have a dominant clone that

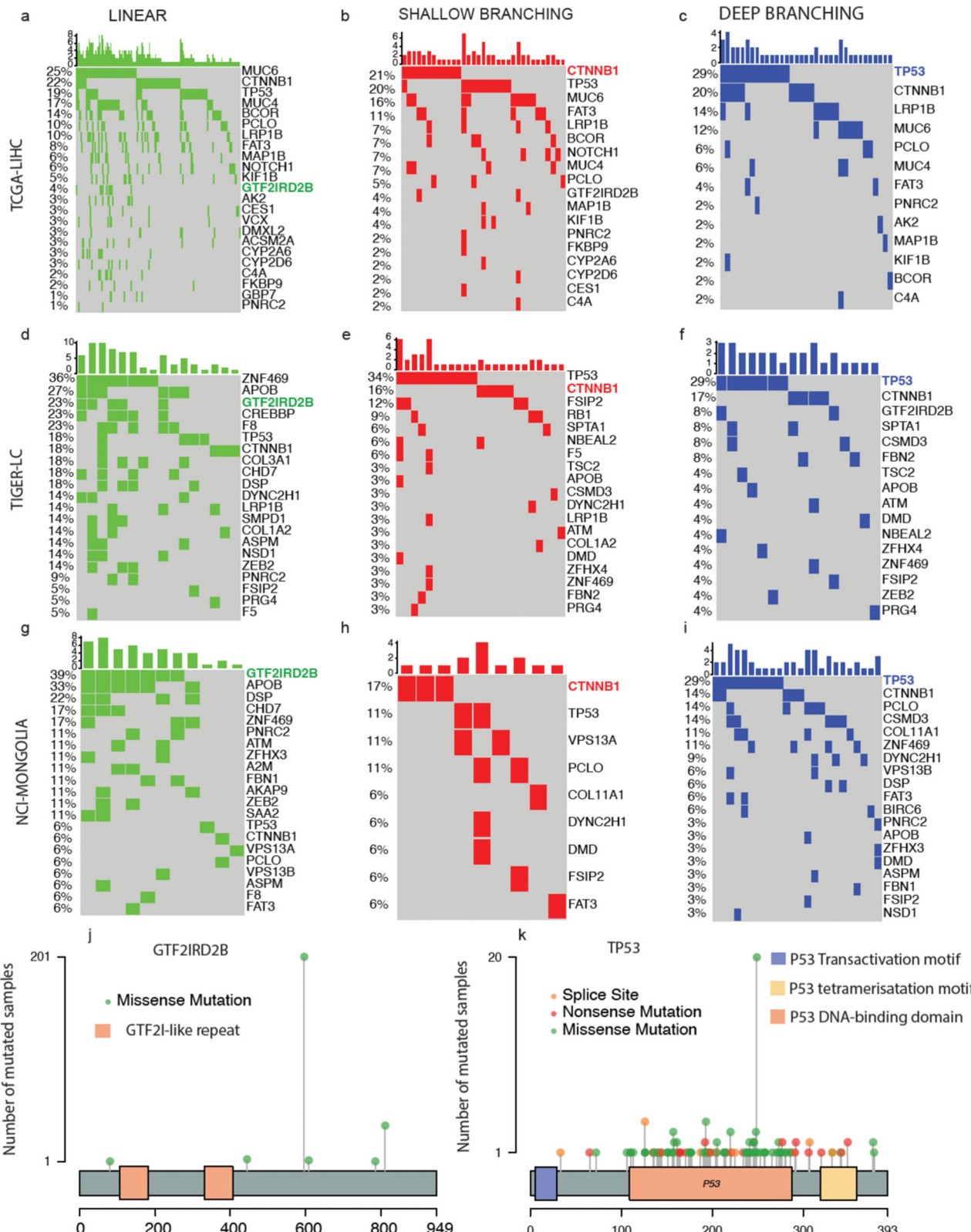

**Fig. 3 | Potential drivers of tumor evolution phylogenies. a–c** Oncoplots showing mutated potential driver genes for linear, shallow branching, and deep branching tree evolution models for TCGA-LIHC cohort. **d–f** Oncoplots showing mutated potential driver genes for linear, shallow branching and deep branching tree evolution models for TIGER-LC cohort. **g–i** Oncoplots showing mutated potential driver genes for l linear, shallow branching and deep branching tree evolution models for NCI-MONGOLIA cohort. The genes in the plots are mutated genes that are in the trunk of tumor clonal trees. Deep branching trees have the highest driver mutation rate for TP53 in all cohorts while for linear trees TP53 is not the highest frequent one in any cohort. **j** Mutated loci on GTF2IRD2B gene are displayed for all patients from TCGA-LIHC, TIGER-LC and NCI-MONGOLIA cohorts. **k** Mutated loci on TP53 gene are displayed for all patients from TCGA-LIHC, TIGER-LC and NCI-MONGOLIA cohorts.

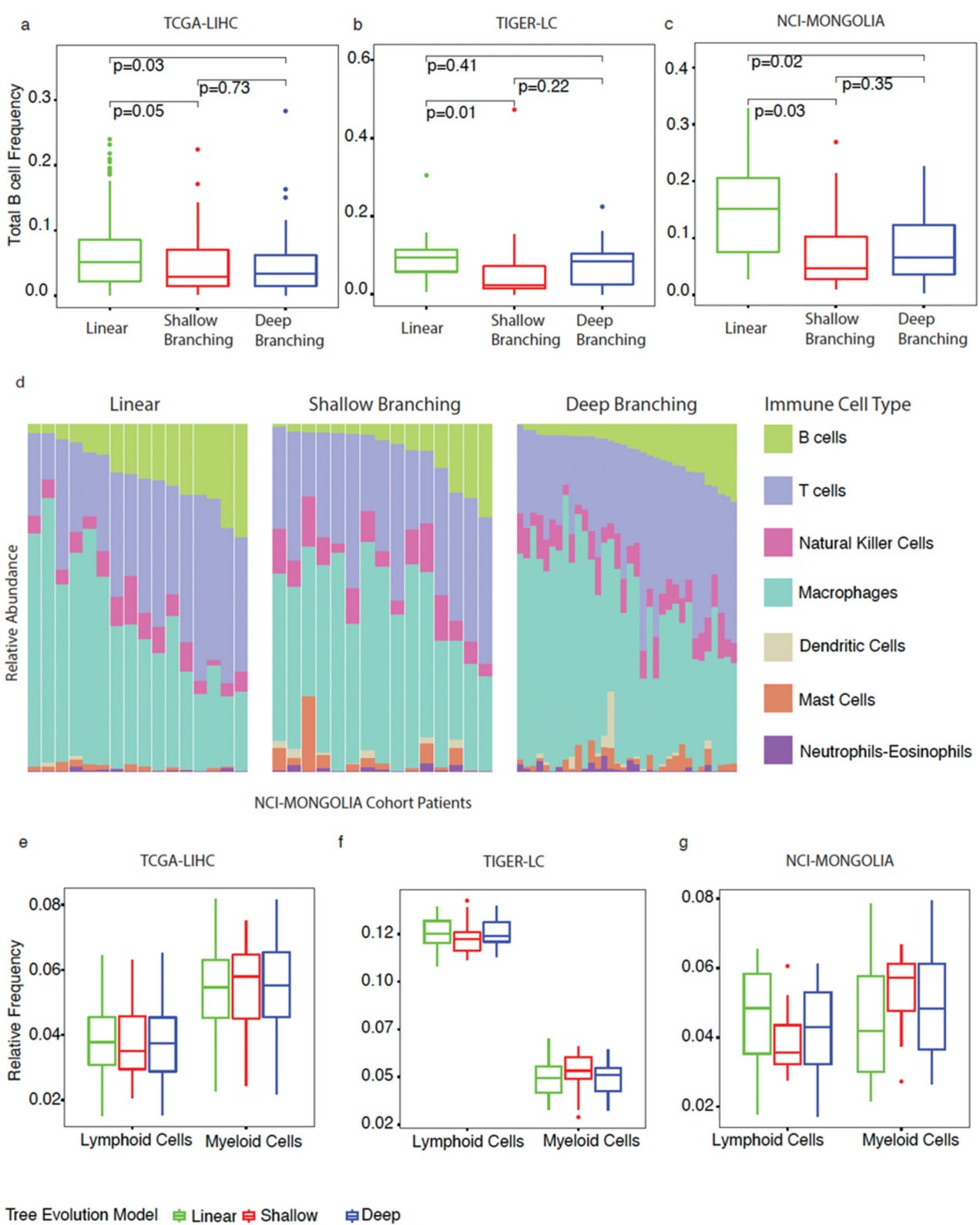

**Fig. 4 | Associating tumor evolution to immune cell microenvironment.**
**a–c** Boxplots showing the total B cell frequencies of linear, shallow branching and deep branching trees for the cohorts TCGA-LIHC, TIGER-LC, and NCI-MONGOLIA respectively. **d** Stacked bar graphs showing the relative frequencies of immune cell types for NCI-MONGOLIA cohort. Color codes for the immune cell type from CIBERSORTX were also provided. **e–g** Grouped boxplots of lymphoid and myeloid cell frequencies by tumor evolution model for the cohorts TCGA-LIHC, TIGER-LC, and NCI-MONGOLIA respectively.

progresses sequentially, and nonlinear trees have multiple major clones that evolve in parallel during tumor progression[31]. This work further classifies nonlinear trees as either deep or shallow branching. Several recent papers[13,31,32] review different tumor evolution models with their distinct biological features including but not limited to linear evolution, branching evolution, neutral evolution, and macroevolution. This study links the deep branching tree to the neutral evolution model, where no selection occurs, resulting in a tree with numerous clones with similar proportions. Therefore, this study corroborates the three well-known tumor evolution models, namely linear, branching, and neutral.

Several studies have shown a correlation between tumor heterogeneity and poor survival[22,33]. Our study confirms this association by demonstrating that high ITH is associated with poor prognosis in cancer since linear trees in our results have the lowest ITH, compared to the shallow and deep branching trees. For all cohorts, the phylogeny type of the tree is associated with survival, so linear trees predict the best survival, and deep branching trees the worst survival.

Some somatic alterations in specific genes, known as "driver genes," contribute to tumorigenesis by granting selective advantages to certain tumor cells[34]. This study aims to find drivers of each tumor phylogeny by examining the founding clone mutations in each phylogeny. The *GTF2IRD2B* gene is a potential driver for a linear phylogeny in hepatocellular carcinoma (HCC), while *TP53* and *CTNNB1* are candidates for driving more aggressive branching phylogenies.

The tumor microenvironment (TME) plays a role together with tumor cells in tumor progression and response to treatment. Successful establishment of tumor clonality requires a comprehensive understanding of the development of somatic alterations in tumor cells and the formation of a conducive TME that facilitates the survival and growth of these altered tumor cells[35]. Thus, the interaction between tumor cells and their microenvironment validates a successful clonality construction. In this study, we aimed to provide evidence for such an interaction. We first focused on B cells in our study because they have been shown to play a crucial role in the immune response against cancer. B cells can produce antibodies that target tumor antigens, leading to their destruction by other immune cells. Additionally, B cells are involved in antigen presentation and immune regulation, which can impact the overall anti-tumor immune response. Our results showed the frequency of B cells is significantly higher in linear trees, which are associated with less aggressive tumors, compared to non-linear trees. This suggests that B cells may play a role in suppressing tumor progression in less aggressive tumors, while they are less effective in more aggressive tumors. These findings are consistent with previous studies that have shown a positive association between the presence of B cells in tumors and improved patient outcomes in HCC[36]. Furthermore, our observation that non-linear trees had the highest myeloid cell frequencies and the lowest lymphoid cell frequencies compared to linear in all three cohorts is also consistent with studies demonstrating a negative correlation between myeloid cells and tumor prognosis[37,38].

One of the limitations of this study is the lack of longitudinal biopsies, as only one tumor sample from each patient was analyzed. This limits the ability to perform clonality analysis using software and tools that are designed for multiple tumor samples. Extending this study to cohorts with multiple tumor samples would strengthen the results by allowing for more comprehensive analysis of clonal evolution over time.

Another limitation of this study is that the results regarding the association between phylogenies and their drivers are purely descriptive. To obtain a causal relationship between these drivers and tumor phylogenies, further experiments involving the genes *GTF2IRD2B*, *TP53*, and *CTNNB1* and their relationship with clonal evolution are needed.

To overcome these limitations, future studies could include longitudinal biopsies and analyze multiple tumor samples from each patient to gain a more comprehensive understanding of clonal evolution. Additionally, functional experiments could be conducted to explore the causal relationship between the identified drivers and tumor phylogenies. Such studies could ultimately enhance our understanding of the underlying mechanisms driving tumor progression and inform the development of more effective therapeutic strategies.

## Methods

### Sample collection and inclusion criteria
TCGA-LIHC: The Cancer Genome Atlas (TCGA) hepatocellular carcinoma (TCGA-LIHC) annotated mutation and RNA sequencing data files for 378 cases were extracted from the GDC portal (https://portal.gdc.cancer.gov/). There were 2 tumor samples for 1 patient and 3 tumor samples for another patient, so we only included 1 tumor sample from each patient to do further analysis. NCI-MONGOLIA: HCC patients were diagnosed via standardized pathology reviews based on the WHO Classification of Tumors (also known as the WHO Blue Books) and via clinical assessments based on CT scans and ultrasound diagnosis. Tumoral and adjacent non tumoral liver tissue samples were collected and frozen at −80 °C after surgical resection at the National Cancer Center in Ulaanbaatar, Mongolia. The study was approved by the Ethics Committee at the National Cancer Center in Ulaanbaatar, Mongolia, and written informed consent was obtained from all participants. TIGER-LC: A set of surgical specimens from 78 HCC patients were used. Patients were diagnosed using combinations of imaging studies, tumor size, level of alpha-fetoprotein (AFP) and histological investigations. Informed consent was obtained from all patients included in this study and approved by the Institutional Review Boards of the respective institutions (NCI protocol number 13CN089; CRI protocol number 18/2555; Chulabhorn Hospital protocol number 11/2553; Thai NCI protocol number EC163/2010; Chiang Mai University protocol number TIGER-LC; Khon Kaen University protocol number HE541099).

### Copy number estimation
Copy number analysis was performed using Sequenza software based on the allele frequencies[16]. Major and minor copy number information, ploidy and tumor purity were estimated and used in SMASH clonality analysis.

### Mutation calling (SNVs), filtering and functional mutations
Mutect2 variant caller was used to identify the mutations from whole exome sequencing data. First, we filtered out the mutations that had less than 7 reads supporting the mutation. Then, we further selected the mutations that are on the specific genes (1006 liver and liver cancer specific genes, Supplementary Data 1). There were 893 unique genes (TIGER-LC (621), NCI-MONGOLIA (625), TCGA-LIHC (820)) out of these functional genes observed in three HCC cohorts used in this study. 386 of those 1006 functional genes were liver-specific genes which were obtained from GTEx normal tissue expression data[11,12]. For those 386 genes, the inclusion criterion was to be expressed at least 10-fold higher in the liver than the median of the 51 non-liver tissues. The remaining functional genes were the candidate drivers of liver cancer extracted from various studies in which authors used methods including integrative pathway crosstalk and protein interaction network, frequency-based approaches, function-based approaches etc.[7,39–41]. Supplementary Fig. 2 shows the hazard ratios, found by exponentiating Cox regression coefficients after allowing for a common unknown duration-specific hazard for 2 groups, linear versus nonlinear.

### Clonality analysis and intra tumor heterogeneity
SMASH was used to perform the clonality analysis[9]. SMASH clusters somatic mutations after estimating copy numbers at mutation locations to identify clones in tumor samples using a likelihood-based framework. All possible phylogenetic trees which were compatible with the data were enumerated while quantifying the probability of each such tree. Given the total number of reads along with the copy number information of each locus, purity of the sample, and the predefined tree structure (linear or nonlinear), the number of mutation reads is modeled by a mixture of binomial distributions. The Expectation-Maximization (EM) algorithm was used to calculate the maximum likelihood estimates in the SMASH algorithm, and the optimal tree of a sample was calculated as the one with the maximum likelihood. For classification of tumor trees, we utilized not only

the proportion entropy, which was already provided in SMASH results, but also defined and calculated the mutation entropy using the probability vector of number of unique mutations (rescaled to sum to 1) in each corresponding subclone. Then, we used the product of these two entropies to define a new metric that measures the intra-tumor heterogeneity (ITH) of a tumor tree. Based on this proposed ITH metric, our data suggested that nonlinear trees had significantly higher intra tumor heterogeneity than linear trees.

## Feature creation for cluster analysis

In the tumor evolution literature, there are several proposed evolution models that may be compatible with resulting nonlinear tumor trees, especially two popular ones. Therefore, we further investigated nonlinear trees to see if there are two subgroups to which we can assign biologically meaningful labels. A clustering algorithm was used to achieve this goal. To simplify our clustering problem, we only searched for two clusters among non-linear tree types. Before applying the clustering algorithm to nonlinear trees, we defined and calculated several features using subclone proportion and number of mutation probability vectors, which we defined as follows. Let $S$ be the total number of subclones in a clonal tree and $q_s$ be the estimated proportion of the $s^{th}$ subclone. Also, denote by $m_s$ as the relative frequency of the unique mutations in the $s^{th}$ subclone. Then, the proportion entropy (PE) and mutation entropy (ME) are defined by:

$$PE = -\sum_{s=1}^{S} q_s \log q_s$$

$$ME = -\sum_{s=1}^{S} m_s \log m_s$$

In addition to these entropies, we created 7 more features, which were given below, by renormalizing the vectors after taking component-wise ratio and product of two entropies, as well as taking squares and cubes of the entries of two entropies. Let $Q = (q_1, q_2, \ldots, q_S)$ be the subclone proportion vector and $M = (m_1, m_2, \ldots, m_S)$ be the mutation frequency vector with $q$ and $m$ being the means of $Q$ and $M$.

$$cor = \frac{\sum_{i=1}^{S}(q_i - q)(m_i - m)}{\sqrt{\sum_{i=1}^{S}(q_i - q)^2(m_i - m)^2}}$$

$$E_{ratio} = -\sum_{i=1}^{s} \frac{\frac{q_i}{m_i}}{\sum_{i=1}^{s}\frac{q_i}{m_i}} \log \frac{\frac{q_i}{m_i}}{\sum_{i=1}^{s}\frac{q_i}{m_i}}$$

$$E_{product} = -\sum_{i=1}^{s} \frac{q_i m_i}{\sum_{i=1}^{s} q_i m_i} \log \frac{q_i m_i}{\sum_{i=1}^{s} q_i m_i}$$

$$ME_{square} = -\sum_{i=1}^{s} \frac{m_i^2}{\sum_{i=1}^{s} m_i^2} \log \frac{m_i^2}{\sum_{i=1}^{s} m_i^2}$$

$$ME_{cube} = -\sum_{i=1}^{s} \frac{m_i^3}{\sum_{i=1}^{s} m^3} \log \frac{m_i^3}{\sum_{i=1}^{s} m_i^3}$$

$$PE_{square} = -\sum_{i=1}^{s} \frac{q_i^2}{\sum_{i=1}^{s} q_i^2} \log \frac{q_i^2}{\sum_{i=1}^{s} q_i^2}$$

$$PE_{cube} = -\sum_{i=1}^{s} \frac{q_i^3}{\sum_{i=1}^{s} q_i^3} \log \frac{q_i^3}{\sum_{i=1}^{s} q_i^3}$$

As a final step in feature creation, we normalized each of the entropy features by log $S$ which is the largest possible entropy with the S number of subclones, in each tumor tree to account for the number of subclone differences (normalized entropy). This choice to normalize by S effectively prevents number of subclones from serving as a predictive feature for clustering, but we also conducted a similar analysis without normalizing by S and found that the unnormalized method produced similar results.

**Cluster analysis of nonlinear trees**. In unsupervised learning methods such as clustering, the data for each record consist of a uniform set of features with no class labels. As a first step, we applied some popular algorithms such as k-means and hierarchical clustering to nonlinear trees with the features described above. However, we did not immediately find an optimal metric to achieve good and stable clustering (under resampling scenarios) that would be consistent for all cohorts. Next, we chose to use a random forest algorithm as a means of establishing a measure of similarity between features more useful than the Euclidian distance even if it was mostly used for supervised learning. Random forests can be used for clustering[42] by considering the original data as class 1 and creating a synthetic second class of the same size that is labeled as class 2. This synthetic class was needed because random forest requires a labeled dataset for training. The synthetic second class is created by sampling at random from the univariate distributions of the original features which destroys the dependency structure among the coordinates of the original data. Thus, the augmented dataset with 2 class labels can be used in the random forest algorithm. In this random forest algorithm, we calculated proximity of a pair of observations (similarity metric) in the following way: grow many random forest trees independently, count the number of times those observations ended up in a same terminal node, and finally normalize this quantity by the number of total trees. Finally, we used this new similarity matrix in k-means algorithm with the following specified settings: centers = 2, nstart = 20, iter.max = 10. The k-means algorithm was implemented with randomly chosen centers and was run many times to get consensus results. Thus, the algorithm extracted 2-class data augmentation 20 times, ran the algorithm 10 times for each data, and chose the optimal clustering as the one with the lowest total sums of squares (distances in all the clusters of the points from the centroids). The proximity-similarity metric used in the k-means algorithm was a nonlinear function of Euclidean distance, as shown in Supplementary Fig. 9. To assign labels to resulting clusters, we used the fact that the deep branching trees are the extreme cases, so we would expect them to have a larger entropy of the terminal node size distribution than shallow branching trees. This labeling strategy is used to align the clusters. For this reason, we calculated the average proportion and mutation entropy of each cluster and found the maximum of these two quantities. Finally, the cluster with the higher maximum was assigned as deep branching. We denote mean mutation entry by ME, proportion entropy by PE, in each case with subscript indicating cluster, and then define

$$m_1 = \max(ME_{cluster1}, PE_{cluster1})$$

$$m_2 = \max(ME_{cluster2}, PE_{cluster2})$$

We label the cluster of trees with larger maximum $m_i$ as deep branching and the other cluster as shallow branching. Principal Component Analysis (PCA) plots were used to visualize the resulting clusters (Supplementary Fig. 6) and confirm their separation.

## Stability analysis of clustering results

For any clustering analysis, it is vital that the clusters were constructed in a stable manner because unstable clustering results might change our biological interpretation. Therefore, we did a stability analysis to show that our shallow and deep branching tree clusters are stable under bootstrap resampling. The data was resampled using nonparametric bootstrapping, i.e.,

the rows (indices) were resampled while keeping the whole observation of fully preprocessed features, and the random forest clustering algorithm was applied to the resampled data. The resulting proximities for each bootstrap-resampled dataset then were used in k-means clustering algorithm. Then, we compared these clustering results of the bootstrap data with the clustering of original data by calculating an affinity score on the 2 × 2 frequency table[22]. We repeated this for B = 1000 bootstrap replications and provided the histogram of estimated affinity scores with p-values in the Supplementary Fig. 10. The following pseudo code summarizes the stability analysis:

For b = 1 to b = 1000:

I. Draw a bootstrap sample $D^*$ of the same size N of original data (randomly select rows from data with replacement).

II. Create another synthetic data $D^{**}$ from $D^*$ by sampling from univariate distributions of features in $D^*$ (each element of a row of $D^{**}$ comes independently from original features).

 a. Run random forest on a problem with 2 labels that represent $D^*$ and $D^{**}$ respectively.

 b. Calculate proximities of a pair of observations of $B^*$. more simply by saying that 1000 random-forest trees were grown on the augmented bootstrap-replicate data ($D^*$, $D^{**}$) and the overall (normalized) Proximity between nodes i,j of $D^*$ is proportional to the count of these trees in which i,j fell in the same cluster.

 c. Aligning the clusters: Feed this proximity matrix to the k-means algorithm and determine the label of each row as shallow or deep branching using $m_1$ and $m_2$.

III. Finally, for each row of $D^*$ compare clustering results for original and bootstrap sample by calculating affinity score (similarity metric) of 2 × 2 frequency table.

## Potential drivers of tumor evolution models

We call a mutated gene a potential driver of the tumor evolution model if it is in the founding clone; in other words, if it is a clonal mutation in the root node of trees of type. We conducted Freeman-Halton extension[24] of the Fisher's exact test for 3 × 3 contingency table to check the association between potential drivers and tumor phylogenies.

## Tumor microenvironment analysis

CIBERSORTx was used to estimate the immune cell decomposition of tumor samples using RNA-seq gene expression data. The gene expression files for the available samples (TCGA (369), TIGER-LC (51), NCI-MONGOLIA (65)) were uploaded to CIBERSORTx as a mixture file, and the analysis was run with the following options: relative and absolute modes together, LM22 signature gene file, 100 permutations and quantile normalization disabled.

## Mutational signature analysis

We used maftools[43] to extract trinucleotide frequency patterns of tumor samples. Reference mutation signatures were obtained from COSMIC (Catalogue of Somatic Mutations in Cancer) version 3 (March 2021). There are a total of 78 reference signatures. An R package deconstructSigs[44] was used to generate mutational signature weights from the nonnegative least squares mapping of individual samples against the reference signatures. After excluding the signatures that are all zero for all patients in the cohort, we ended up with 43 signatures for TCGA-LIHC, 15 signatures for TIGER-LC and 14 signatures for NCI-MONGOLIA.

Random permutation test for the significant differences between mutation signatures of linear versus non-linear trees was implemented. The null hypothesis states that randomly assigning tree labels and taking the mean difference between these new groups is the same as the difference between the original groups. P-values calculated by comparing the rank of the original difference in permuted differences. For each signature, we implemented a random permutation test. The weights for the signature SBS15 which is associated to defective DNA mismatch repair were significantly higher in linear trees compared to shallow or

deep branching in TCGA-LHC and NCI-MONGOLIA cohorts, but we did not see any significant result for TIGER-LC cohort. Supplementary Data 2 provides the p-values of resulting permutation tests for each signature for each cohort.

## Reporting summary

Further information on research design is available in the Nature Portfolio Reporting Summary linked to this article.

## Data availability

The processed whole exome sequencing data. The publicly available datasets used in this study include the TCGA database (TCGA-LIHC) [https://portal.gdc.cancer.gov/projects/TCGA-LIHC]. For NCI-MONGOLIA cohort, RNA Sequencing data are available at the Gene Expression Omnibus (GEO) repository under Study Accession GSE144269, and Whole-Exome Sequencing data are available at the dbGaP repository under Study Accession phs002000.v1.p1. TIGER-LC data available at the dbGap repository under Study Accession phs001199.v2.p1.

## Code availability

Code is available upon request. It should be directed to and will be fulfilled by the Lead Contact, Xin Wei Wang (xw3u@nih.gov).

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

## Acknowledgements
This work was supported by grants (Z01 BC 010877, Z01 BC 010876, Z01 BC 010313, and ZIA BC 011870) from the intramural research program of the Center for Cancer Research, National Cancer Institute of the United States to X.W.W. Z.K. was supported by the UMD-NCI Partnership for Integrative Cancer Research. J.C. was supported by the Intramural Research Program of the National Institute on Aging, NIH.

## Author contributions
Z.K., X.W.W., and E.S. developed study concept; Z.K. performed data analysis; A.R., Jittiporn. C., J.S., and M.F. performed sample collection and processing; Z.K., E.S., D.L., Julian C. and X.W.W. interpreted data; Z.K. wrote the manuscript. A. B., X. W., B.T., J.S., Y.P., Jittiporn. C., S.R., B.P., V.B., N. L., C.A., S.S., C.M., M.R, X.W.W. read, edited, and approved the manuscript.

## Funding

## Competing interests
The authors declare no competing interests.
