## [Peer Review File · Communications Biology]

Reviewers' comments:

Reviewer #1 (Remarks to the Author):

Kacar and colleagues present a study focusing on defining tumour evolution paths in HCC. By using SMASH approach, they reconstructed tumour clonal trees in three independent HCC cohorts. Interestingly, only when focusing on selected functional genes, they observed a difference in survival, particularly patients characterised by linear evolution showed a better survival. They then attempted to link the clonality results with TME composition and found the frequency of B cells being higher in linear trees compared to non-linear trees. Finally, an analysis of mutational signatures revealed no differences in mutational exposure profiles between linear and non-linear trees.

Overall the work is of interest but I have some suggestions below that could help to clarify some aspects of the analysis and help the reader to understand better the results:

1. In line 123, the authors state that the NCI-MONGOLIA cohort has a smaller sample size ($n=71$) compared to TCGA-LIHC ($n=375$) and TIGER-LC ($n=78$), so for this reason they focused their analysis on these last two cohorts. Could the author explain why do they keep the TIGER-LC cohort in their analysis when the number of patients is very similar to NCI-MONGOLIA cohort? In addition, it's not very clear that they focused only in two cohorts, as they showed all three cohorts in most of the figures.

2. In line 177-180, the authors state that the TCGA-LIHC cohort consisted mostly of linear trees (270 linear, 56 shallow branching, 49 deep branching), while the TIGER-LC (22 linear, 32 shallow branching, 24 deep branching) and NCI-MONGOLIA (18 linear, 18 shallow branching, 35 deep branching) cohorts had a higher proportion of shallow and deep branching trees. Did the authors consider the stage? It seems to me that the TCGA cohort is particularly enriched in early stage. Could the authors show these results subdividing the cohorts by early and late-stage?

3. The survival analysis showing that the deep branching tumours are more aggressive is interesting and in line with previous findings. Have the authors performed a multivariate analysis (including factors such as stage, gender, age) to assess if this result is really driven by IHT?

4. The analysis on comparing driver genes in the three different phylogenies needs a more validated approach to define a gene as "driver". I would suggest tools such as "dNdScv" (Martincorena et al., Cell 2017) or "MutSigCV" (Lawrence et al., Nature 2013). In addition, it's not clear if for this analysis they focused on the selected genes (as they did in the previous analysis) or they looked at all genes.

Minor comments:

1. Supplementary figures 1a and b are in fact figure 1b. I would remove them from the supplementary file as they are redundant.

2. Supplementary figure 2 also included TIGER-LC cohort but there is no mention in the figure legend.

3. In line 209-212 the authors claim "Although boxplots in Figures 2j, 2k, and 2l showed that linear trees had reasonably normal ploidy estimates (around 2)

compared to non-linear trees, ploidy was not found to be a significant factor in distinguishing evolutionary tree types". However they don't show any data, nor do they explain how they performed such analysis. More clarity is needed here.

Reviewer #2 (Remarks to the Author):

The authors of this paper attempted to characterize tumor evolution through the analyses of cancer driver gene mutation profiles in three cohorts of liver cancer patients, but with the following problems:

1. Please indicate whether the basic information and disease background of the three HCC patients are consistent, which is crucial for the applicability of the conclusions of this paper to HCC of a specific etiology.
2. In the Supplementary Figure 2, the prognostic results of the TIGER-LC cohort are not statistically different and are inconsistent with those mentioned in the article. It cannot be concluded from the TIGER-LC cohort that functional mutations are more effective in distinguishing patients with different prognosis.
3. From the view of clinical application value, can we stratify patients with HCC into the subtypes mentioned in this work? How to do so?
4. Regarding the correlation analysis of the evolutionary tree with immune cell infiltration, I am concerned that from the results. B cells are indeed proportionally higher in the linear model. However, in Figure 4e-f, we cannot visually compare the differences between lymphocytes and myeloid cells in each subtype due to the lack of P-values. Also, there seems to be no difference in the proportion of immune cells infiltrating in the linear model and the deep branching subtype, and I suggest that the authors revise their conclusions.
5. For the prognostic analysis of the three subtypes, I suggest that the authors show the P values for a two-by-two comparison (figure 2d-f). Graphically, the prognostic differences between the linear model and Shallow B do not seem to be significant.

Reviewer #1 (Remarks to the Author):

Kacar and colleagues present a study focusing on defining tumour evolution paths in HCC. By using SMASH approach, they reconstructed tumour clonal trees in three independent HCC cohorts. Interestingly, only when focusing on selected functional genes, they observed a difference in survival, particularly patients characterized by linear evolution showed a better survival. They then attempted to link the clonality results with TME composition and found the frequency of B cells being higher in linear trees compared to non-linear trees. Finally, an analysis of mutational signatures revealed no differences in mutational exposure profiles between linear and non-linear trees.

Overall the work is of interest but I have some suggestions below that could help to clarify some aspects of the analysis and help the reader to understand better the results:

1. In line 123, the authors state that the NCI-MONGOLIA cohort has a smaller sample size (n=71) compared to TCGA-LIHC (n=375) and TIGER-LC (n=78), so for this reason they focused their analysis on these last two cohorts. Could the author explain why do they keep the TIGER-LC cohort in their analysis when the number of patients is very similar to NCI-MONGOLIA cohort? In addition, it's not very clear that they focused only in two cohorts, as they showed all three cohorts in most of the figures.

Thank you for your question, and we appreciate your attention to the details of our study.

We began our investigation by comparing "all mutations versus functional mutations" within the TIGER-LC cohort. We then aimed to validate our findings using two additional HCC cohorts that are available to us. However, our "all mutations versus functional mutations comparison" was conducted only on the TIGER-LC and TCGA-LIHC cohorts, and not on the NCI-Mongolia cohort, for reasons of cohort comparability, as can be seen in the new Supplementary Table 3. This table compares key demographic factors of age, gender distribution, and cancer stage among the tumor samples across the different cohorts. The NCI-MONGOLIA cohort exhibited a notably higher proportion of late-stage cancer patients compared to the other cohorts, and its follow-up times were notably shorter. For these reasons, we excluded NCI-MONGOLIA cohort from our initial analysis. The inclusion of the TCGA-LIHC cohort in our analysis was driven by its substantial sample size (n=371), which added statistical robustness to our comparison alongside the TIGER-LC cohort. As for the appearance of all three cohorts in some of our figures, this was a deliberate choice made to provide a comprehensive overview of our findings after we did the functional mutation analysis on NCI-MONGOLIA cohort as well. So, we did functional clonality analysis for all three cohorts and presented the results but did not implement "all versus functional comparison" for NCI-MONGOLIA cohort. In response to your comment, we have now refined the text in lines 126-135 of the main manuscript to ensure greater clarity regarding our analytical approach. We trust that this explanation sheds light on our methodology and the rationale behind our cohort selection.

2. In line 177-180, the authors state that TCGA-LIHC cohort consisted mostly of linear trees (270 linear, 56 shallow branching, 49 deep branching), while the TIGER-LC (22 linear, 32 shallow branching, 24 deep branching) and NCI-MONGOLIA (18 linear, 18 shallow branching, 35 deep branching) cohorts had a higher proportion of shallow and deep branching trees. Did the authors consider the stage? It seems to me that the TCGA cohort is particularly enriched in early stage. Could the authors show these results subdividing the cohorts by early and late stage?

Thanks for your comment. Yes, we did see that TCGA-LIHC cohort has many more linear trees compared to the other two cohorts. We did consider the stage of cancer when analyzing the data. However, despite the differences in tree type distribution among the cohorts, the statistical tests we applied did not detect a significant within-cohort relationship between tree type and cancer stage. We did see a significant association when all three cohorts are combined. However, in the TIGER-LC cohort the stage information is often missing. In some cases, the numbers of patients with specific combinations of tree type and stage are quite small, which limits the statistical power to detect significant associations. With pooled data from all cohorts, there is some evidence to suggest that the distribution of phylogenetic tree types may vary between early and late stages of cancer. This could potentially indicate a role for phylogenetic tree type in cancer progression. However, we are hesitant to over-interpret these results, in light of the heterogeneity of the cohorts and the low statistical power of the analysis.

We did a complete case analysis in analyzing the stage variable. However, stage was missing for 45 out of 78 cases in TIGER-LC cohort, 17 out of 71 cases in NCI-MONGOLIA, and 24 out of 375 cases for TCGA-LIHC cohort as shown in supplementary table 1. That is why especially for smaller cohorts the results of complete-case analysis will not be reliable.

Further investigation on additional cohorts may be necessary to confirm and better understand the nature of this association. Additionally, considering the non-significant results in the separate cohort analyses, the association between these variables may be cohort-specific or influenced by other factors not captured in the current analysis.

We added the following tables as **Supplementary Tables 7-9**. We also mention these tables briefly in the **text lines 232-236**.

Supplementary table 7. Test for Independence of Phylogeny Type vs. Cancer Stage for TCGA-LIHC cohort: Pearson's Chi-squared Test (2 df) for 3 x 2 Contingency Table.

TCGA-LIHC	Early	Late
Linear	195	61
Shallow Branching	36	16
Deep Branching	30	13
	Pearson`s Chi-squared test p-value=0.44 for row-column independence	

Supplementary table 8. Test for Independence of Phylogeny Type vs. Cancer Stage for TIGER-LC cohort: Fisher`s Exact test for 3x2 Contingency Table.

TIGER-LC	Early	Late
Linear	7	2
Shallow Branching	5	4
Deep Branching	11	4
Fisher`s exact test p-value=0.63		

Supplementary table 9. Test for Independence of Phylogeny Type vs. Cancer Stage for NCI-MONGOLIA cohort: Fisher`s Exact test for 3x2 Contingency Table.

NCI-MONGOLIA	Early	Late
Linear	4	9
Shallow Branching	5	7
Deep Branching	10	19
Fisher`s exact test p-value=0.84		

3.The survival analysis showing that the deep branching tumors are more aggressive is interesting and in line with previous findings. Have the authors performed a multivariate analysis (including factors such stage, gender, age) to assess if this result is really driven by IHT?

Thank you for your comment. Firstly, because of the heavily missing values in the stage variable, the complete case analysis we used could be a biased approach considering that we do not know and cannot model the nature of these missing values. Still, to address your concern, we have performed multivariate analysis. To assess whether the observed result indicating that deep branching tumors are more aggressive is driven by Intratumor Heterogeneity (IHT) while controlling for other potential factors such as stage, gender, and age, a survival analysis was performed using Cox Proportional-Hazards Regression with multiple predictors.

TCGA-LIHC: Analysis of Deviance Table (Type II tests)		
N=350 (117 dead)	LR Chisq	P-value
Tree-type	9.87	0.0077
Stage	17.84	<0.0001
Age	3.23	0.0722
Gender	0.05	0.8318

The partial likelihood Type-II test is calculated to test each term in the presence of all others. Since Tree-type variable is significant even after controlling for age, stage, and gender, this strengthens the conclusion that it is associated with survival outcome.

For TIGER-LC and NCI-MONGOLIA cohorts, unfortunately we had many missing values. That is why the findings of such test in these cohorts are very limited. We still emphasize that the combination of complete-case analysis and the possible failure of the proportional-hazards assumption makes this Cox-model analysis only suggestive rather than definitive. We added this table for TCGA-LIHC cohort along with 3 more tables as Supplementary Tables 10-13 and mentioned these limited results in the main text line 237-246.

4. The analysis on comparing driver genes in the three different phylogenies needs a more validated approach to define a gene as “driver”. I would suggest tools such as “dNdScv” (Martincorena et al., Cell 2017) or “MutSigCV” (Lawrence et al., Nature 2013). In addition, it’s not clear if for this analysis they focused on the selected genes (as they did in the previous analysis) or they looked at all genes.

Thanks for your comment and suggestions. Regarding the analysis comparing driver genes in the three different phylogenies, we appreciate your recommendations to employ tools such as "dNdScv" (Martincorena et al., Cell 2017) or "MutSigCV" (Lawrence et al., Nature 2013) for a more robust assessment of driver genes. We acknowledge the merit of these tools for cases with multiple samples for each patient. Unfortunately, in our study, we were constrained by having only one tumor sample per patient, which limited our ability to utilize such tools effectively.

You raised a valid point about the clarity of our analysis. We would like to clarify that we focused on selecting functional cancer-specific genes that from which all potential drivers are likely drawn. Within this context, we restricted attention to mutations that fell within the first clone of each phylogenetic tree. While we do not definitively label them as drivers, they are potential drivers for each specific phylogeny type. We now emphasized these in several places in the main text line 99, 263-264.

These mutations are of particular importance as they represent the initiating events in the tumorigenesis process.

Minor comments:

1. Supplementary figures 1a and b are in fact figure 1b. I would remove them from the supplementary file as they are redundant.

Thank you for bringing this to our attention, and we appreciate your feedback. We have removed these redundant figures from the supplementary file.

2. Supplementary figure 2 also included TIGER-LC cohort but there is no mention in the figure legend.

Thank you for pointing out, we fixed that one, too.

3. In line 209-212 the authors claim "Although boxplots in Figures 2j, 2k, and 2l showed that linear trees had reasonably normal ploidy estimates (around 2) compared to non-linear trees, ploidy was not found to be a significant factor in distinguishing evolutionary tree types". However, they don't show any data, nor do they explain how they performed such analysis. More clarity is needed here.

Thank you for your comment. We did the comparison analysis for ploidy estimates and did only see significant difference between ploidy estimates of different phylogenies for NCI-MONGOLIA cohort. We now clearly state the analysis in the Figure 2 (j-l) caption and mentioned this in the main text line briefly in lines 250-256.

Reviewer #2 (Remarks to the Author):

The authors of this paper attempted to characterize tumor evolution through the analyses of cancer driver gene mutation profiles in three cohorts of liver cancer patients, but with the following problems:

1. Please indicate whether the basic information and disease background of the three HCC patients are consistent, which is crucial for the applicability of the conclusions of this paper to HCC of a specific etiology.

Thank you for your valuable comment. We understand the significance of ensuring the consistency of basic information and disease background across our study cohorts, particularly concerning the applicability of our conclusions to specific etiologies of HCC. We have addressed your comment as follows. In addition to the previously provided Supplementary Table 1, which contains some cohort characteristics, we have now included further descriptive and comparative information about the cohorts in Supplementary Table 3. We also mentioned this table and limitation in main text lines 130-132. This new table provides a more comprehensive comparison, including the means of the age variable as well as the differences in gender and stage proportions among the cohorts.

2. In the Supplementary Figure 2, the prognostic results of the TIGER-LC cohort are not statistically different and are inconsistent with those mentioned in the article. It cannot be concluded from the TIGER-LC cohort that functional mutations are more effective in distinguishing patients with different prognosis.

We sincerely appreciate your thoughtful observation regarding the prognostic results of the TIGER-LC cohort in Supplementary Figure 2 and its inconsistency with the information presented in the article. It is indeed crucial to distinguish between statistical significance and practical significance in data analysis. A p-value slightly above the conventional threshold of 0.05 should not dismiss the potential presence of meaningful trends or associations within the data, especially when compared to significantly higher p-values. This insight highlights the

importance of not relying solely on arbitrary p-value thresholds but also considering the effect size and the broader context of the analysis.

Regarding the specific p-value of 0.08 obtained in the functional analysis, we underlined that it compares linear versus nonlinear mutations within the TIGER-LC cohort. However, throughout this paper, we are firstly interested in testing a hypothesis of no survival difference between linear versus nonlinear tree types in all cohorts versus alternatives where all cohorts show linear versus nonlinear effects in the same direction. To test this, we specifically calculated a cohort stratified log-rank test. Strikingly, this test unveiled a highly significant p-value of 0.00001, indicating a strong association between the tree-type variable (linear or nonlinear) and survival. We mentioned these in the main text lines 145-151.

3. From the view of clinical application value, can we stratify patients with HCC into the subtypes mentioned in this work? How to do so?

Thank you for your inquiry and interest in the clinical application of our work. It is important to note that the results presented in our study are suggestive rather than assertive, and we appreciate your consideration of their clinical implications.

Stratifying patients with HCC into the subtypes discussed in our study is indeed an intriguing prospect. In the Methods section of our paper does provide an algorithmic method for clustering the patients in these cohorts into phylogeny groups, but that an analogous algorithm for genomic assignment of new patients to phylogeny groups would require a large amount of training and validation data. Although the findings of statistical significance suggest some biological persistence of the conclusions, that is not guaranteed without considerable additional testing on a larger variety of patients, with greater attention to protocols of accrual and recording of baseline characteristics such as cancer stage, before the findings could be confidently generalized to clinical practice. Therefore, we would like to emphasize that further research and validation are essential before any concrete clinical applications can be developed. The subtypes we have identified may provide a foundation for future investigations, but the process of translating these findings into clinical practice involves several critical steps.

4. Regarding the correlation analysis of the evolutionary tree with immune cell infiltration, I am concerned that from the results. B cells are indeed proportionally higher in the linear model. However, in Figure 4e-f, we cannot visually compare the differences between lymphocytes and myeloid cells in each subtype due to the lack of P-values. Also, there seems to be no difference in the proportion of immune cells infiltrating in the linear model and the deep branching subtype, and I suggest that the authors revise their conclusions.

We appreciate your input, and we would like to address your concerns. The lack of statistically significant differences in the proportion of immune cells infiltrating between the linear model and the deep branching subtype is noted. We understand your desire for P-values to provide a quantitative measure of significance in Figure 4e-f. Our intention in presenting those figures was to visually convey potential trends and insights, rather than assert definitive statistical significance. Again, our aim in presenting these figures was to stimulate further investigation and

exploration in the scientific community, and we welcome further analysis and research in this area. In the light of the lack of statistically significant differences in immune cell infiltrations across phylogeny groups, we have revised the main text to say that these B-cell exhibits are merely descriptive and provide nothing more than a suggestion that further research may discover differences in immune cell types in genomically determined phylogeny types. You can find those revised **text in lines 58-62, 103-105, and some text in lines 300-307.**

5. For the prognostic analysis of the three subtypes, I suggest that the authors show the P values for a two-by-two comparison (figure 2d-f). Graphically, the prognostic differences between the linear model and Shallow B do not seem to be significant.

Thank you for your suggestion regarding the prognostic analysis of the three subtypes. You are correct in noting that the log-rank test p-values for the comparison between the linear model and Shallow B did not reach the threshold for statistical significance.

We again would like to emphasize that the hypothesis that we are testing is no survival difference between linear versus nonlinear tree types versus alternative that the differences all have the same sign. In response to your suggestion, we also have incorporated the p-values for the two-by-two comparison of these subtypes into our analysis. We have provided these pairwise survival curve comparison p-values in Supplementary Table 5. We also include the stratified (by cohort) log-rank test p-values for those pairwise comparisons in Supplementary Table 4.

With the clonality analysis results, we had significant linear versus nonlinear phylogenies, which agreed with previous findings in the literature. The novelty in this paper was to find subgroups in the non-linear category. We successfully showed that our clustering strongly separates the shallow and deep branching groups within the nonlinear category, in a systematic way across cohorts. However, as you realized, survival outcomes for the shallow phylogenies are hardly worse than those for the linear ones, despite this separation. One argument to strengthen the case for their difference was to investigate how the structural characteristics of the shallow and linear trees impact various aspects of the data. We compared the number of total mutations in linear versus shallow branching tree types and found out that the linear trees have significantly lower number of mutations compared to Shallow Branching (Supplementary Table 6). We have now added the **text in lines 178-188 of** the main manuscript to make these findings clearer.

Supplementary table 4. Stratified Log Rank Test P-Values for Comparing Pairwise Phylogeny Clusters Stratified by Cohort. Each stratified log rank chi-square, assessing pairs of phylogeny groups treated as two-sample data stratified by cohort, is based on 1 degree of freedom.

	All cohorts stratified
Linear-Shallow	Chisq=1.3, p-value=0.3
Shallow-Deep	Chisq=10.4, p-value=0.001

Deep-Linear	Chisq=16.7, p-value<0.001
Linear-Nonlinear	Chisq=13.1, p-value<0.001

Supplementary table 5. P-Values from Log Rank Tests Comparing Pairwise Phylogeny Clusters Across Cohorts. All p-values are calculated using 1-degree-of-freedom log rank chi-squares. These log rank tests are derived from standard two-group survival statistics, each based on pairs of phylogeny-group data treated as two-sample datasets.

Cohort	Linear vs Shallow Branching	Linear vs Deep Branching)	Shallow Branching vs Deep Branching	Linear vs Nonlinear (Deep and Shallow Branching)
TCGA-LIHC	0.51	<0.001	0.007	0.03
TIGER-LC	0.57	0.009	0.04	0.08
NCI- MONGOLIA	0.3	0.4	0.9	0.3

Supplementary table 6. 2-Sample Welch t-Test for Comparing Mean Mutations Between Linear and Shallow Branching Phylogeny Populations, Assuming Independence. Test statistic for the comparison of Linear vs Shallow Branching Tree Types for TCGA-LIHC, TIGER-LC, and NCI-MONGOLIA are -2.72, -2.02, and -1.74 respectively.

	TCGA-LIHC	TIGER-LC	NCI-MONGOLIA
Linear vs Shallow Branching	<0.001	0.04	0.09